# Diversity of Ophiostomatoid Fungi Associated with *Dendroctonus armandi* Infesting *Pinus armandii* in Western China

**DOI:** 10.3390/jof8030214

**Published:** 2022-02-22

**Authors:** Huimin Wang, Tiantian Wang, Ya Liu, Fanyong Zeng, Haifeng Zhang, Cony Decock, Xingyao Zhang, Quan Lu

**Affiliations:** 1Key Laboratory of Forest Protection of National Forestry and Grassland Administration, Institute of Forest Ecology, Environment and Nature Conservation, Chinese Academy of Forestry, Beijing 100091, China; wanghuimin@caf.ac.cn (H.W.); sstr94@163.com (T.W.); 13436862712@163.com (Y.L.); xyzhang@caf.ac.cn (X.Z.); 2Chinese Academy of Forestry, Beijing 100091, China; zeng@caf.ac.cn; 3Longcaoping Forestry Bureau of Shaanxi Province, Foping County, Hanzhong 723401, China; zhangxinyi0302@163.com; 4Mycothèque de l’Université Catholique de Louvain (BCCM/MUCL), Earth and Life Institute, Microbiology, B-1348 Louvain-la-Neuve, Belgium; cony.decock@uclouvain.be

**Keywords:** *Esteyea*, *Graphilbum*, *Graphium*, *Leptographium*, *Ophiostoma*, taxonomy

## Abstract

*Pinus armandii* (*P. armandii*) is extensively abundant in western China and, as a pioneer tree, and prominently influences local ecology. However, pine forests in this region have been significantly damaged by *Dendroctonus armandi* (*D. armandi*) infestations, in close association with ophiostomatoid fungi. This study aimed to identify the diversity of ophiostomatoid fungi associated with *D. armandi * infesting *P. armandii* in western China. A total of 695 ophiostomatoid fungal strains were isolated from 1040 tissue pieces from *D. armandi* galleries and 89 adult beetles at four sites. In this study, based on multiloci DNA sequence data, as well as morphological and physiological characteristics, seven species belonging to five genera were identified including three known species, *Esteyea vermicola*, *Graphium pseudormiticum* and *L. wushanense*, two novel taxa, *Graphilbum parakesiyea* and *Ophiostoma shennongense*, and an unidentified *Ophiostoma* sp. 1. A neotype of *Leptographium qinlingense*. *Ophiostoma shennongense* was the dominant taxon (78.99%) in the ophiostomatoid community. This study provides a valuable scientific theoretical basis for the occurrence and management of *D. armandi* in the future.

## 1. Introduction

The ubiquitous yet diverse associations between insects and fungi have long evolved [1,2,3], while the interactions between beetles, microbes, and hosts have been well documented [4,5,6,7,8,9,10], showing a variety of ecological strategies [11,12,13]. The associations among *Pinus* spp., *Dendroctonus* spp. (*Coleoptera*, *Curculionidae*, *Scolytinae*), and ophiostomatoid fungi are among of the most significant types of hosts-beetles-mycobiota mutualism.

*Pinus armandii* (*P. armandii*) is a native and pioneer coniferous species on the Qinling Mountains in China [14] that has a significant role in local economy and ecology. However, *P. armandii * has been infested with *Dendroctonus armandi* (*D. armandi*) since it was first reported in 1932 [15]. *D. armandi* is an endemic beetle species that gregariously attacks 20- to 50-year-old healthy, *P. armandii* mainly in western China [16,17]. To date, in an area spanning more than 4000 ha, approximately half a million *P. armandii* trees have been decimated by this beetle [18,19].

The bark beetle-associated mycobiota, particularly ophiostomatoid fungi (*Ophiostomatales*, *Microascales*, *Ascomycota*), has been extensively studied due to its diversity, pathogenicity, and mutualism [20,21]. Previous studies have demonstrated that most fungi associated with *D. armandi* belong to *Alternaria*, *Trichoderma*, *Verticillium*, and ophiostomatoid fungi [18,22,23]. To date, ten species have been assigned to the order Ophiostomatales (*Leptographium qinlingensis*, *L. terebrantis*, *Leptographium* sp., *Leptographium* sp1., *Leptographium* sp2., *Ophiostoma brevicolle*, *O. floccosum*, *O. quercus*, and *Ophiostoma* sp.), and Microascales (*Ceratocystis polonica*) [18,22,23,24,25]. However, the records of *L. terebrantis* and *C. polonica* are uncertain due to the lack of confirmatory molecular analysis reports [22,24]. The former is associated with many bark beetles that are only recorded in North America, while the latter is associated with *Ips* spp. infesting spruce [26,27,28].

Research on the diversity of ophiostomatoid fungi associated with *D. armandi* on *P. armandii* in China remains limited. There are no systematic studies, with only a few sporadic ophiostomatoid fungi reports [23,24,25]. Furthermore, the status of *L. qinlingensis* has been challenged, due to the absent type specimen, molecular analysis and limited morphological characteristics to prove that *L. qinlingensis* was a new species [24]. Therefore, the validity of *L. qinlingensis* should be considered if similar material is obtained from the same vector and host [29].

In this study, we explored the ophiostomatoid communities associated with *D. armandi* infecting *P. armandii* ecosystems in western China. Integrated morphological observations and multilocus DNA sequence data were used to analyze these communities. Our results provide insights into the communities of ophiostomatoid fungi associated with *D. armandi* in western China, which is a basic assignment for the subsequent study on the occurrence and management of *D. armandi*.

## 2. Materials and Methods

### 2.1. Sample Collection and Fungi Isolation

Samples including *D. armandi* adults and their breeding galleries were collected from infected *P. armandii* trees at four sites (Table 1) in western China from July to August 2018 and May to July 2019 (Figure 1). All four sites are pure forests of *P. armandii* with tree ages of approximately 40 years old and diameters of approximately 40 to 60 cm. The trees used in this study showed signs of being dead or dying. The beetles were individually placed in sterilized Eppendorf tubes using tweezers, while their galleries were placed in sterile envelopes using a sterilized knife. Beetles and galleries were returned to the laboratory and stored at 4 °C for isolation within one week.

Beetles were crushed directly without superficial disinfection and transferred to a 2% malt extract agar (MEA: 20 g Biolab malt extract, 20 g Biolab agar, and 1000 mL deionized water). Galleries were cut into smaller tissue sections (5 × 5 mm), disinfected with 1.5% sodium hypochlorite (NaClO) for 60 s, rinsed with sterile water three times, and placed in 9 cm petri dishes, as described by Seifert et al. [30]. All strains were purified using mycelium apex, and cultures were grown in the dark at 25 °C. According to the preliminary analysis of culture characteristics, representative strains of each morphotype were selected for further morphological and molecular studies. All fungal strains obtained in this study were maintained in the culture collection of the Chinese Academy of Forestry (CXY), and representative strains were maintained in the China Forestry Culture Collection Center (CFCC, part of the National Infrastructure of Microbial Resources) (Table 2).

### 2.2. Morphological and Physiological Characteristics

Pure cultures were incubated in the dark at 25 °C, culture morphology and growth status were observed daily, and the microstructures of reproduction forms were performed on 2% MEA media and incubated for 7 to 30 days. Microscope slides were prepared to observe the length and width of reproductive structures (such as conidiogenous apparatus, stipes cylindrical, conidiophore, and conidia) per strain using a BX51 OLYMPUS microscope with differential interference contrast. In total, 30 measurements were repeated for each morphological feature, and the statistics were presented as (min–) (mean-SD) –(mean + SD) (–max) (mean, average; SD, standard deviation; min, minimum; max, maximum).

For growth rate studies, representative strains were cultured in 90 mm diameter plates in the dark. A total of five replicate plates were included for each strain incubated at 5 °C intervals (5 to 35 °C) for two weeks. The diameter of each colony was measured daily until the mycelium reached the edge of the MEA medium. Colony colors were described according to Rayner’s color chart [31].

### 2.3. DNA Extraction and Sequencing

Before DNA extraction, the strains were grown on 2% MEA for 1–2 weeks at 25 °C in the dark. The mycelia of purified strains were picked from the 60 mm diameter plates and placed into 2 mL sterile Eppendorf tubes. DNA extraction and purification were performed using the Plant Genomic DNA Kit (Invisorb Spin Plant Mini Kit, DP305, Tiangen, Beijing, China), following the manufacturer’s protocol.

A total of five DNA regions were amplified for sequencing and phylogenetic analyses. The internal transcribed spacer regions (ITS1 and ITS2, including the 5.8S gene) were amplified using the ITS1/ITS4 primer pair [32]; the nuclear ribosomal large subunit region (LSU) was amplified using the LROR/LR5 primer pair [33]; ITS2 and part of the ribosomal large subunit 28S (ITS2-LSU) were amplified using the ITS/LR3 primer pairs [32]; the β-tubulin (TUB2) gene was amplified using the BT2a/BT2b primer pair [34]; and the elongation factor1-α (EF1-α) gene was amplified using the EF1F/EF2R primer pair [35]. PCR reactions were conducted in 25 μL volumes (2.5 mM MgCl_2_, 1× PCR buffer, 0.2 mM dNTP, 0.2 mM of each primer, and 2.5 U Taq-polymerase enzyme), and PCR amplification was conducted using a thermocycler (Applied Biosystems, Foster City, CA, USA). The reaction conditions for the five DNA regions were similar to those described in the references for primer design. PCR products were cleaned with an MSB Spin PCR Apace Kit (250) following the manufacturer’s instructions. All nucleotides were sequenced in both directions using a CEQ 2000 XL capillary automated sequencer (Beckman Coulter), and MEGA5.0 was used for splicing.

### 2.4. Phylogenetic Analyses

Preliminary identification of the obtained sequences based on ITS DNA fragments was performed using BLAST searches in the NCBI GenBank database. The related authentic sequences were downloaded for further phylogenetic analyses. Sequence alignment was performed online using MAFFT (http://mafft.cbrc.jp/alignment/server/ accessed on 13 December 2021), implementing the iterative refinement method (FFT-NS-i setting) [36], and edited with MEGA5.0. The gaps were treated as the fifth base. Maximum likelihood (ML), maximum parsimony (MP), and Bayesian inference (BI) were used to assess these aligned sequences in phylogenetic analysis.

ML analyses were conducted using RAxML v. 7.0.3, [37] under the GTR-GAMMA model. Supports for the nodes were estimated from 1000 bootstrap replicates. MP analyses were performed using PAUP*version 4.0b10 [38]. A bootstrap analysis (1000 replicates using the neighbor-joining option) was performed to determine the support levels of the nodes. BI analyses were conducted using MrBayes v. 3.1.2 [39]. A total of four Markov chain Monte Carlo (MCMC) chains were run simultaneously from a random starting tree for five million generations, and samples were taken per 100 generations, resulting in 50,000 trees. Moreover, the first 25% of trees sampled were discarded during burn-in. Posterior probabilities were calculated by the retained trees. The topology of the resulting files was subsequently visualized using Figtree v.1.4.2 and Adobe Illustrator CS6.

## 3. Results

### 3.1. Fungal Isolation

In this study, 695 strains of ophiostomatoid fungi were isolated from 1040 tissue pieces from *D. armandi* galleries and 89 adult beetles at four sites (Table 2 and Table 3). A total of 274 strains (39.42%) and 421 strains (60.58%) were isolated from the beetles and their galleries, respectively. Moreover, 280, 69, and 346 strains were obtained from the Hubei, Gansu, and Shaanxi provinces, respectively (Table 3).

Growth rates, colony characteristics, and ITS sequence BLAST results were used for preliminary sorting and identification. These strains were distributed across five genera (*Esteya*, *Graphilbum*, *Leptographium*, *Ophiostoma* in the Ophiostomatales, and *Graphium* in the Microascales) and seven tentative species/groups (Taxon 1 to 7; Table 2). A total of 30 representative strains were selected for subsequent morphological and phylogenetic analyses.

### 3.2. DNA Sequencing and Phylogenetic Analyses

Phylogenetic analyses of the ITS, ITS2-LSU, LSU, TUB2, and TEF1-α gene regions were used to identify the genera/species and the genetic diversity within ophiostomatoid fungi [29,40,41]. Overall, one to eight strains were selected for each tentative species (Taxa 1–7) to construct the phylogenetic trees. The topologies generated by the ML, MP, and BI analyses were highly concordant, and phylograms obtained by ML were presented for all individual datasets, with branch supports obtained from MP and BI analyses.

*Esteya* (Taxon 1) and *Graphium* (Taxon 3) were each represented by a single strain. The two combined datasets (LSU+TUB2 and LSU+TEF1-α) for Taxon 1 and Taxon 3 consisted of 1198 and 1114 characters (including gaps), respectively, grouped with *Esteya vermicola* and *Graphium pseudormiticum* (Appendix A).

The remaining 28 representative strains were distributed in three major clades by phylogenetic inferences based on the ITS and ITS-LSU data sets, including *Ophiostoma s. str*, *Leptographium s. l*, and *Graphilbum s. str*.

Taxon 2 was represented by three (Table 2) of the 25 strains (Table 3). It formed a well-supported clade in both ITS and TUB2 based phylogenies, closely related to *Graphilbum kesiyea* but distinct (Figure 2 and Figure 3); hence its recognition as a distinct species.

Taxon 4 included 62 strains (Table 3), while Taxon 5 included 34 strains (Table 3), with 11 representative strains (Table 2) belonging to the *Leptographium lundbergii* complex in the ITS2-LSU phylogenetic analyses (Figure 4). Additionally, eight strains (Table 2) of Taxon 4 formed a distinct and well-supported clade based on the TUB2 dataset (Figure 5). However, ITS2-LSU and TEF1-α based phylogenetic inferences revealed that these eight strains clustered together with *L. qinlingensis* with high support (Figure 4 and Figure 6). There was no TUB2 sequence for *L. qinlingensis* prior to this. *L. qinlingensis* is a *nom. inval*., because of the lack of type specimen.

Taxon 5 grouped with Taxon 4 in the *L. lundbergii* complex and three representative strains of Taxon 5 grouped with *L. wushanense*, as defined by Pan et al. [42], based on ITS2-LSU and TUB2 phylogenetic analyses (Figure 4 and Figure 5).

Taxon 6 included 549 strains (Table 3), seven of which (Table 2) were included in the phylogenetic analyses and clustered in the *Ophiostoma clavatum* complex. A total of seven (Table 2) of the 22 strains (Table 3) of Taxon 7 resided outside any recognized species complexes (Figure 2). The analysis of ITS and TUB2 sequences (Figure 2 and Figure 7) showed that Taxon 6 formed a well-supported clade close to, yet distinct from, the ex-type sequences of *O. clavatum*. Meanwhile, Taxon 7, nested in the vicinity of *O. aggregatum*, forming a clade related to, yet distinct in the ITS and TUB2 phylogenetic trees (Figure 2 and Figure 8). Hence, these strains in Taxon 6 and Taxon 7 were interpreted as belonging to two distinct, undescribed *Ophiostoma* species.

### 3.3. Morphology and Taxonomy

A total of three of the seven taxa identified in the present study were interpreted as undescribed species. These included one species of *Graphilbum* (Taxon 2) and two species of *Ophiostoma* (Taxa 6 and 7). However, no reproductive structures were observed for Taxon 7 on the different media used in this study; thus, we have elected to refrain from describing this taxon at this time. *L. qinlingense* was recollected during this study, and the name is revalidated by designation of a Neotype.

Taxonomy.

*Graphilbum parakesiyea* T. Wang & Q. Lu sp. nov. Figure 9.

MycoBank: MB838526.

Etymology: ‘*parakesiyea*’ (Latin), refers to the phylogenetic affinities to *Graphilbum kesiyea*.

Type: China, Hubei Province, galleries of *Dendroctonus armandi* in *Pinus armandii*, Aug. 2018, TT Wang, holotype CXY2516, ex-type CFCC53924 = CXY2516.

Description: Sexual state not observed.

*Asexual morph*: hyalorhinocladiella-like. Conidiophores arising directly from the mycelium, simple or loosely branched, reduced to conidiogenous cells; conidiogenous cells aseptate or sparsely septate, thin-walled, with a rounded apex, hyaline (17.8–) 32.2–80 (–116.3) × (1.2–) 1.6–2.3 (–2.8) μm. Conidia hyaline, single-celled, aseptate, smooth, clavate to elliptical with obtuse ends, (4.6–) 4.9–5.6 (–6.2) × (1.9–) 2.0–2.7 (–3.2) μm.

Culture Characteristics: Colonies grew rapidly on 2% MEA, attaining 90 mm days at 30 °C in the dark, while, with appressed hyphae, colonies are white with a smooth margin. The optimum growth temperature is 30 °C; however, it can grow from 5 °C to 35 °C.

Known substrate and hosts: Galleries and adults of *Dendroctonus armandi* in *Pinus armandii*.

Known insect vectors: *Dendroctonus armandi*.

Known distribution: Hubei and Shaanxi provinces, China.

Additional specimens examined: CHINA, Shaanxi Province, Foping country, galleries, and adults of *Dendroctonus armandi* in *Pinus armandii*, May to July 2019, TT Wang, CFCC 54514 = CXY2539, CFCC 54515 = CXY2540.

Notes: *Graphilbum parakesiyea* is characterized by hyalorhinocladiella-like asexual morph. Phylogenetic analysis (Figure 2 and Figure 3) revealed that *Gra. parakesiyea* is closely related to *Gra. kesiyea*. *Graphilbum parakesiyea* can be distinguished from *Gra. kesiyea* by their conidiogenous cells; conidiogenous cells of *Gra. kesiyea* are longer than that of *Gra. parakesiyea*, *viz.* 38–101.5 μm and 32.2–80 μm, respectively. The two species also differ in their optimal growing temperatures of 25 °C and 30 °C, respectively [43]. Furthermore, *Gra. kesiyea* was isolated from *Pinus kesiya* infected by *Polygraphus aterrimus* and *Polygraphus szemaoensis* [43], while *Gra. parakesiyea* was isolated from *P. armandii* infected with *D. armandi*.

*Leptographium qinlingense* (M. Tang) T. Wang & Q. Lu comb. nov. Figure 10.

≡ *Ophiostoma qinlingensis* Tang, journal of Huazhong Agricultural University 23:5. 2004.

Type: no specified. Neotype: China, Shaanxi Province, galleries of *Dendroctonus armandi* in *Pinus armandii*, June 2019, TT Wang, neotype designated here CFCC53941 = CXY2515.

MycoBank: MB838528.

Description: Sexual state not observed.

*Asexual morph*: leptographium-like and hyalorhinocladiella-like.

Leptographium-like. Conidiophores erect, macronematous, mononematous, arising directly from the mycelium, (100.5–) 115.9–219.1 (–302.8) μm long, differentiated into a stipe and a conidiogenous apparatus. Stipes cylindrical, straight, 1–4 septate, constricted at septa, brown to dark brown, (17.1–) 27.5–109.2 (–203.7) μm long, (1.8–) 3.3–6.3 (–7.8) μm in diameter; conidiogenous apparatus (14.0–) 25.6–66.7 (–104.4) μm long, with 2–3 series of cylindrical branches; primary branch cylindrical, pale brown, smooth, (12.0–) 14.4–35.8 (–46.1) × (2.1–) 3.1–4.7 (–6.1) μm; conidiogenous cells cylindrical, discrete, hyaline, (6.0–) 10.2–22.6 (–32.1) μm in length, (1.5–) 2.1–4.0 (–5.4) μm wide; conidia holoblastic, hyaline, single-celled, aseptate, oblong to obovoid, clavate, (5.3–)7.2–11.7 (–14.7) × (3.3–)3.9–5.9 (–6.8) μm.

Hyalorhinocladiella-like: conidiophores arising directly from the mycelium, simple branched, macronematous, or semi-macronematous, mononematous, the ultimate branched bearing conidiogenous cells; conidiogenous cells septate, hyaline, think-walled, rounded apex, (21.4–) 26.5–78.1 (–105.8) × (1.23–) 1.3–2.7 (–4.5) μm; conidia hyaline, smooth, single-celled, aseptate, elliptical to obovoid with truncate base and rounded apex, (3.2–) 4.5–7.7 (–8.8) × (1.8–) 2.4–4.4 (–5.16) μm.

Culture characteristics: Colonies grew rapidly on 2% MEA, attaining a 90 mm diameter after five days at 25 °C in the dark, accounting for a daily growth rate up to 20 mm. Colonies have a smooth margin, radial hyphae, curved shape, initially hyaline, discoloration progressing to olivaceous from the center of the colonies to the margin. The optimum growth temperature is 25 °C; however, growth can occur from 5 °C to 35 °C.

Known substrate and hosts: Galleries and adults of *Dendroctonus armandi* in *Pinus armandii*.

Known insect vectors: *Dendroctonus armandi*.

Known distribution: Shaanxi, Gansu and Hubei provinces, China.

Additional specimens examined: CHINA, Gansu Province, Dangchun Forest Farm, galleries and adults of *Dendroctonus armandi* in *Pinus armandii*, August 2018, TT Wang, CFCC 53937 = CXY2510, CFCC53938 = CXY2511, CFCC 53923 = CXY2512, CFCC 53939 = CXY2513, CFCC 53940 = CXY2514; CHINA, Shaanxi Province, Foping County, galleries and adults of *Dendroctonus armandi* in *Pinus armandii*, May to July 2019, TT Wang, CFCC 54521 = CXY2541, CFCC 54516 = CXY2542.

Notes: *Leptographium qinlingense* was first isolated from *D. armandi* which infects *P. armandii* in China [24]. In this study, the strain CFCC53941 was isolated from exactly the same vector and host as the first reported of *L. qinlingense*. Thus, it was designated as the neotype herein. No differences were observed in the cultures or morphological characteristics between the recently collected neotype and that in the first reported article. Measurements of the asexual structures were consistent with previous descriptions of *L. qinlingense* (Figure 10).

*Leptographium qinlingense* is characterized by a leptographium-like and hyalorhinocladiella-like asexual morphs. It is closely related to *G. koreana*, *L. pinicola*, and *L. truncatum* based on ITS2-LSU, TUB2, and TEF1-α genetic phylogeny (Figure 5 and Figure 6). Within the *L. lundbergii* complex, *L. qinlingense* is the sole species containing two asexual morphs (Figure 10). *Grosmannia koreana* is the sole species with a sexual morph [44]. Moreover, the four species differ in terms of the leptographium-like morph spore size. Specifically, the conidia of *L. qinlingense* (4.5–7.7 μm) are longer than that of *L. pinicola*, (3–5 μm) and either longer than *G. koreana* (3–10 μm), yet shorter than *L. truncatum* (7–11 μm) [44,45,46]. While *L. qinlingense* is native and endemic to China’s mainland, the remaining three species of the *L. lundbergii* complex are widely distributed in the North temperate hemisphere (Canada, China, England, Korea, and the USA) and the southern hemisphere (New Zealand and South Africa). Furthermore, these four species are associated with bark beetle infestation of conifers [45,46,47].

*Ophiostoma shennongense* T. Wang & Q. Lu sp. nov. Figure 11.

MycoBank: MB838527.

Etymology: ‘shennongense’ (Latin), referring to the locality.

Type: China, Hubei Province, galleries of *Dendroctonus armandi* in *Pinus armandii*, Aug. 2018, TT Wang, holotype CXY2501, ex-type CFCC53921 = CXY2501.

Description: Sexual state not observed.

*Asexual morph*: hyalorhinocladiella-like. Conidiophore arising directly from mycelium, simple branched, macronematous or semi-macronematous, mononematous, the ultimate branched bearing numerous conidiogenous cells; conidiogenous cells hyaline, smooth, thin-walled, aseptate, rounded apex, variable in length, (7.1–) 30.0–82.3 (–110.9) × (1.2–) 1.4–2.1 (–2.6) μm; conidia holoblastic, hyaline, single-celled, smooth, aseptate, elliptical to obovoid with truncate base and rounded apex, (4.4–) 5.7–7.4 (–8.0) × (2.0–) 2.2–3.0 (–3.5) μm.

Culture characteristics: Colonies grown on 2% MEA, attaining a 70 mm diameter after eight days at 25 °C in the dark, while, with appressed hyphae, colonies are white with a smooth margin, discoloration progresses to pale olivaceous from the center of the colonies to the margin. The optimal temperature is 30 °C; however, growth can begin from 5 °C to 35 °C.

Known substrate and hosts: Galleries and adults of *Dendroctonus armandi* in *Pinus armandii*.

Known insect vectors: *Dendroctonus armandi*.

Known distribution: Hubei, Shaanxi, and Gansu provinces, China.

Additional specimens examined: CHINA, Hubei Province, Shennongjia Forest Area, galleries and adults of *Dendroctonus armandi* in *Pinus armandii*, August 2018, TT Wang, CFCC 53922 = CXY2502; CHINA, Gansu Province, Dangchun Forest Farm, *Dendroctonus armandi* galleries and adults in *Pinus armandii*, August 2018, TT Wang, CFCC 53931 = CXY2503, CFCC 54528 = CXY2534; Shaanxi Province, Foping county, galleries and adults of *Dendroctonus armandi* in *Pinus armandii*, May to July, 2019, TT Wang, CFCC 53932 = CXY2505; Shaanxi Province, Huoditang Forest Farm, galleries and adults of *Dendroctonus armandi* in *Pinus armandii*, May to July, 2019, TT Wang, CFCC 54534 = CXY2535, CFCC 54533 = CXY2536.

Notes: The sole reproductive structure of *O. shennongense* formed on 2% MEA was a hyalorhinocladiella-like morph. *Ophiostoma shennongense* belongs to *O. clavatum* complex (Figure 7) [48]. The sexual stages of this complex were characterized by brown, spirally coiled ostiolar, hyphae, and cylindrical-to-rectangular ascospores. The asexual stages are hyalorhinocladiella-like to pesotum-like. *Ophiostoma shennongense* is closely related to *O. clavatum* based on ITS and TUB2 phylogenetic analyses (Figure 2 and Figure 7). These species differ by their colony color and hyalorhinocladiella-like conidial size. The conidia of *O. shennongense* (5.7–7.4 μm) are larger than in *O. clavatum* (4–5μm) [48,49]. *Ophiostoma shennongense* colonies are pale olivaceous, whereas the colonies of *O. clavatum* are dark brown to almost black [48].

## 4. Discussion

This study was undertaken to determine the diversity of ophiostomatoid fungi associated with *D. armandi* infesting *P. armandii* in the Qinling Mountains of western China. A total of 695 strains of ophiostomatoid fungi were identified from seven species in five genera comprising of four known species, *E. vermicola*, *Gra. pseudormiticum*, *L. wushanense*, and *L. qinlingense*, as well as a novel neotype strain, as assessed by its type, locality, and combination insect/host (CFCC53941); two novel taxa, *Gra. parakesiyea*, *O. shennongense*; and an unidentified *Ophiostoma* sp. 1.

Among the seven species of ophiostomatoid fungi, *O. shennongense* was the most frequently isolated species, accounting for an abundance of 78.99%, representing the predominant component of the community associated with *D. armandi-P. armandii* (Table 3), compared to the second most abundant species, *L. qinlingense* (8.92%). *Leptographium qinlingense* was the first ophiostomatoid species reported to be associated with *D. armandi*, and is currently isolated only from China [24]. This species has previously been shown to exhibit pathogenicity with high virulence [18,25,50]. Due to its common occurrence associated with *D. armandi* infesting *P. armandii*, *L. qinlingense* may have a significant role in the damage observed in *D. armandi*-infected *P. armandii* in China [22,50,51,52].

This study is the second report showing that *L. wushanense* associates with *P. armandii*; however, *P. armandii* is infected by *D. armandi* in Shaanxi Province rather than by *Tomicus armandii* in Yunnan Province [42]. Before this study, only one strain of *L. wushanense* has been reported from *T. armandii*, which showed occasional association with the beetle and pine. In the present study, *L. wushanense* was also isolated only at Huoditang Forest Farm out of four investigated sites, showing a limited occurrence (Table 3). The species, therefore, are sporadically located throughout southwestern China and loosely associated with the beetle.

*Esteya vermicola* and *Gra. pseudormiticum* were each represented by a single strain. *Esteya* is a unique genus of Ophiostomataceae, with two species: *E. vermicola* and *E. floridanum*. Both species exhibit high infectivity toward the pinewood nematode (*Bursaphelenchus xylophilus*) by their lunate conidia, and are potentially biocontrol agents against this epidemic pine disease [53,54,55,56,57,58]. *Esteya vermicola* was first isolated from Japanese black pine in Taiwan in 1999, and is associated with the pinewood nematode [53]. Since then, eight strains have been recorded worldwide [53,54,55,56,57,58]. However, although the species appear to be widely distributed, only a single strain was recorded in each of these previous studies. *Graphium pseudormiticum* was first reported in South Africa and was associated with *Orthotomicus erosus* [59], subsequently reported in Sweden as associated with *Ips typographus*, in Austria associated with *Tomicus minor* [60], and in China associated with *Pissodes* sp., a mite of *Ips acuminatus* [43,61].

Although an association between fungi and bark beetles has been observed, the classic theory of reciprocal symbiosis between bark beetles and fungi has been challenged due to the lack of in-depth explanation of the symbiosis mechanism or the existence of contradictory research cases. The present study expanded our knowledge of *D. armandi* and its associated fungi; however, the symbiosis mechanism between ophiostomatoid fungi and *D. armandi* warrant further investigations.

## Figures and Tables

**Figure 1 jof-08-00214-f001:**
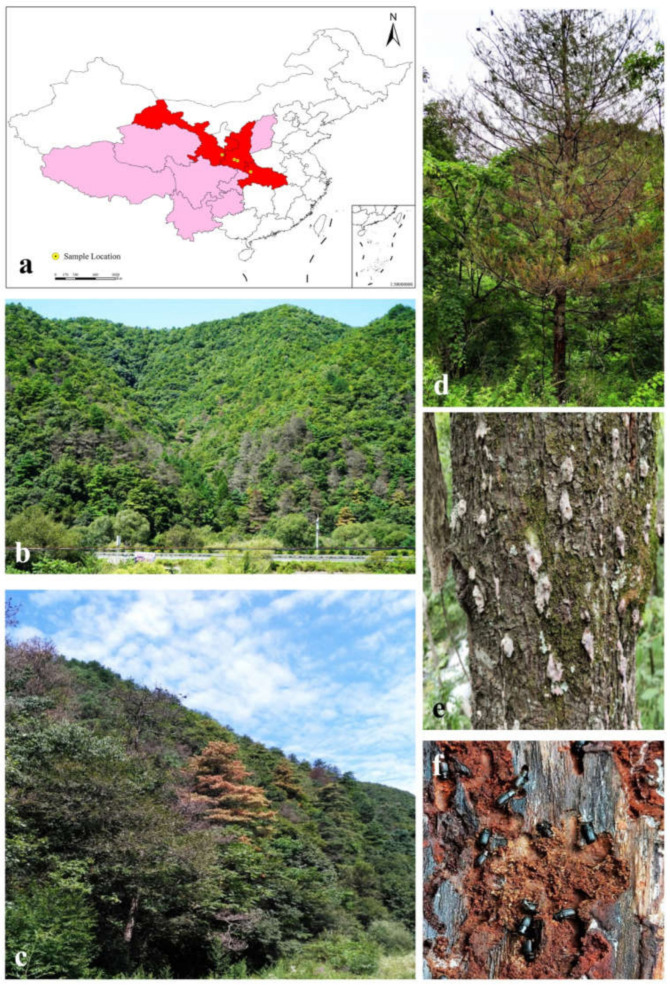
(**a**) Map showing the distribution of *P. armandii* and sample locations of *D. armandi* in China. (**b**–**e**) Disease symptoms of *P. armandii* infested by *D. armandi* and ophiostomatoid fungi in the Qinling Mountains of western China. (**f**) Adult *D. armandi* in galleries on *P. armandii*.

**Figure 2 jof-08-00214-f002:**
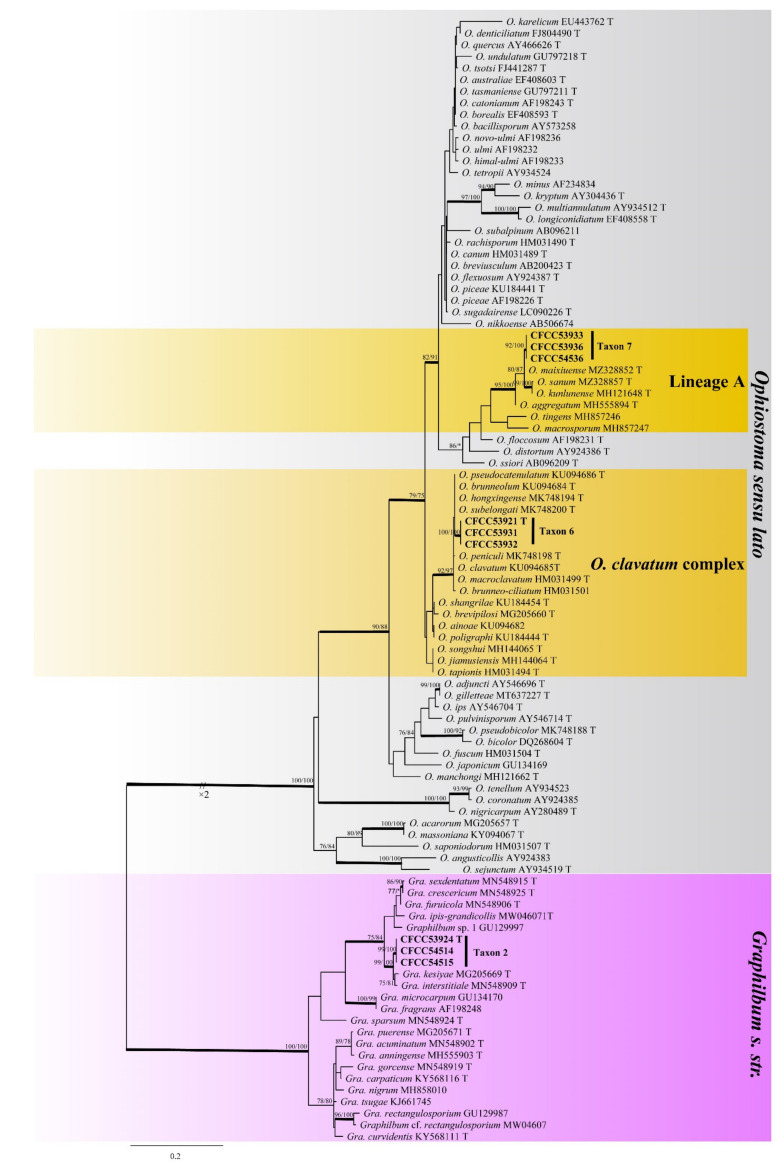
ML tree of *Ophiostoma sensu lato* and *Graphilbum s. str*. (Taxa 2, 6, and 7) generated from the ITS sequence data. Novel sequences obtained in this study are presented in bold typeface. Bold branches indicate posterior probability values ≥ 0.9. Bootstrap values ≥ 70% for ML and MP are indicated above branches. Bootstrap values < 70% are indicated by *. Strains representing ex-type sequences are marked with ‘T’.

**Figure 3 jof-08-00214-f003:**
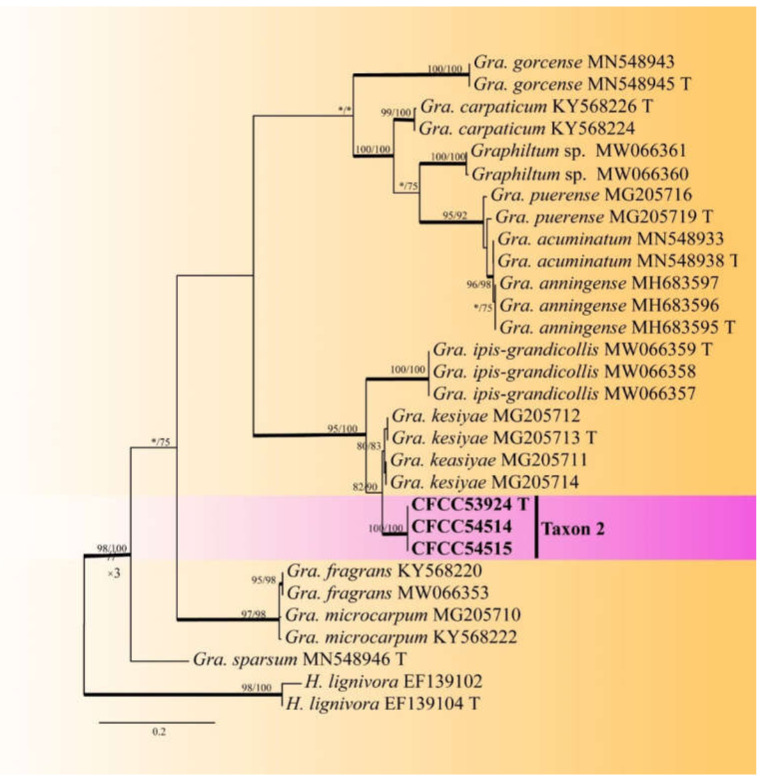
ML tree of *Graphilbum s. str.* generated from the TUB2 sequence data. Novel sequences obtained in this study are presented in bold typeface. Bold branches indicate posterior probability values ≥ 0.9. Bootstrap values ≥ 70% for ML and MP are indicated above branches. Bootstrap values < 70% are indicated by *. Strains representing ex-type sequences are marked with ‘T’.

**Figure 4 jof-08-00214-f004:**
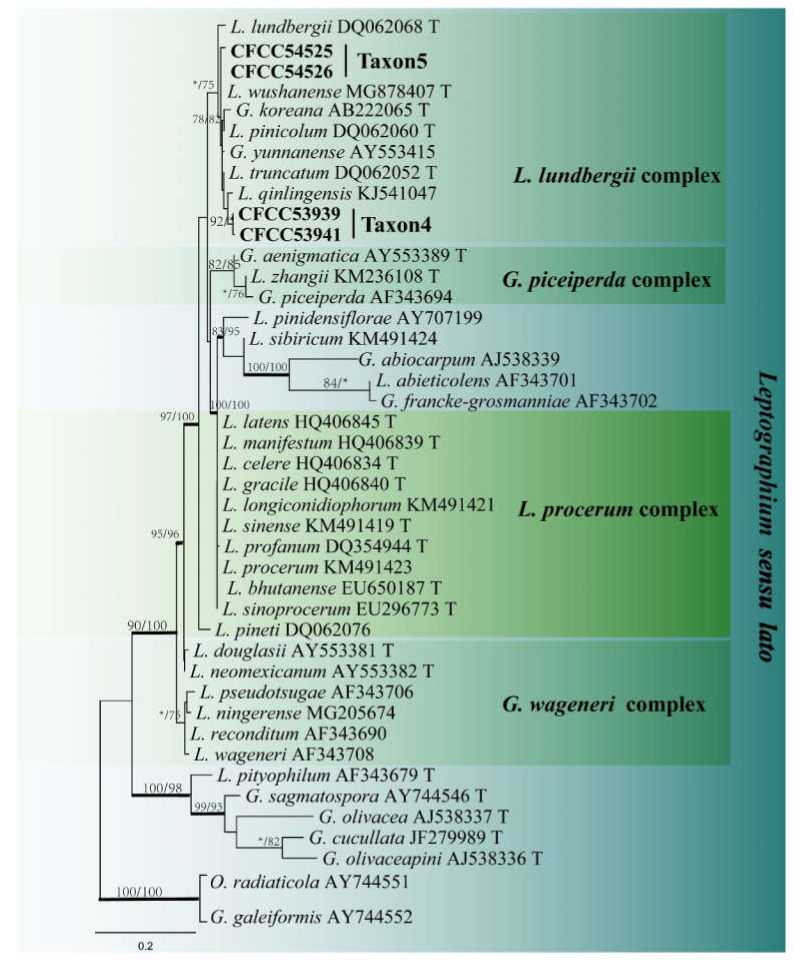
ML tree of *Leptographium sensu lato* (Taxa 4 and 5) generated from the ITS2-LSU sequence data. Novel sequences obtained in this study are presented in bold typeface. Bold branches indicate posterior probability values ≥ 0.9. Bootstrap values ≥ 70% for ML and MP are indicated above branches. Bootstrap values < 70% are indicated by *. Strains representing ex-type sequences are marked with ‘T’.

**Figure 5 jof-08-00214-f005:**
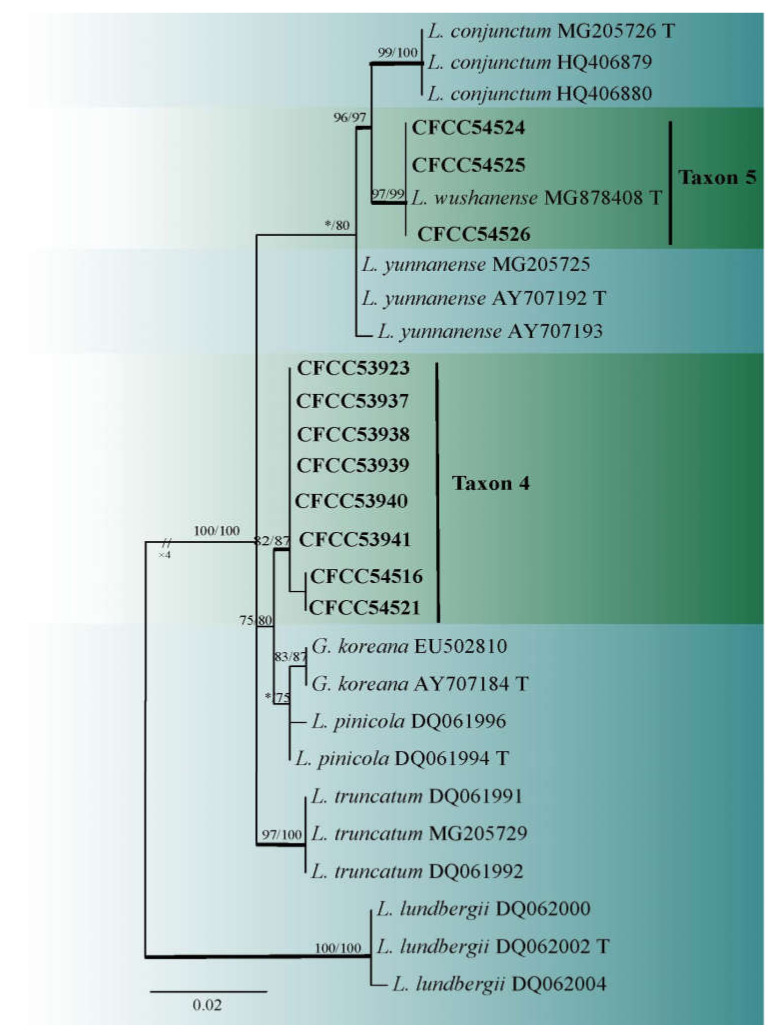
ML tree of *L. lundbergii* complex (Taxa 4 and 5) generated from the TUB2 sequence data. Novel sequences obtained in this study are presented in bold typeface. Bold branches indicate posterior probability values ≥ 0.9. Bootstrap values ≥ 70% for ML and MP are indicated above branches. Bootstrap values < 70% are indicated by *. Strains representing ex-type sequences are marked with ‘T’.

**Figure 6 jof-08-00214-f006:**
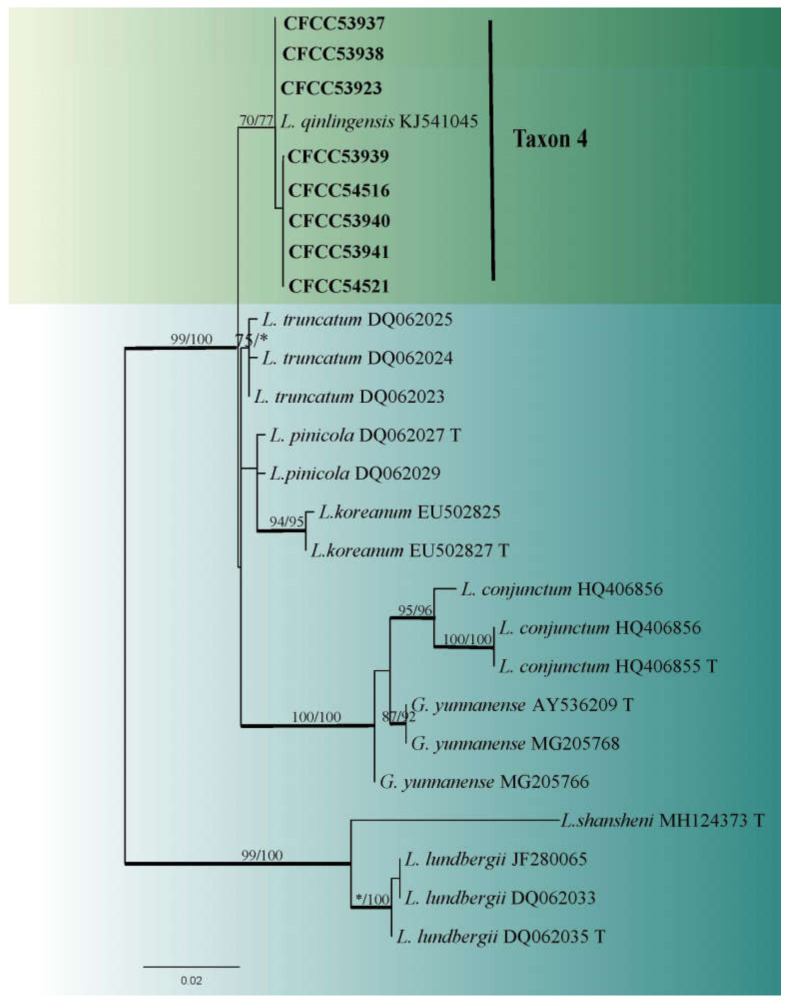
ML tree of *L. lundbergii* complex (Taxon 4) generated from the TEF1-α sequence data. Novel sequences obtained in this study are presented in bold typeface. Bold branches indicate posterior probability values ≥ 0.9. Bootstrap values ≥ 70% for ML and MP are indicated above branches. Bootstrap values < 70% are indicated by *. Strains representing ex-type sequences are marked with ‘T’.

**Figure 7 jof-08-00214-f007:**
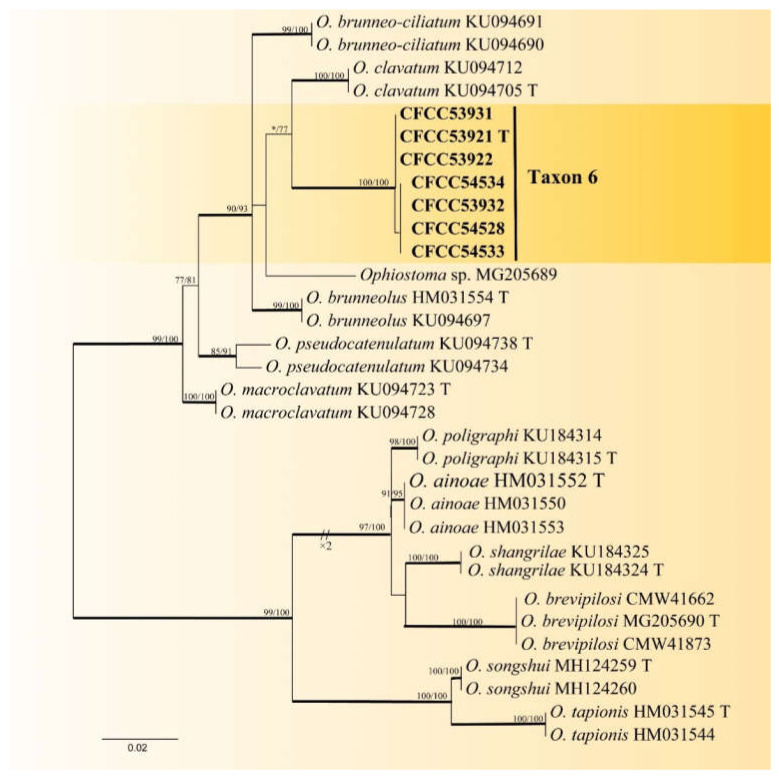
ML tree of *O. clavatum* complex (Taxon 6) generated from the combined (ITS + TUB2) sequence data. Novel sequences obtained in this study are presented in bold typeface. Bold branches indicate posterior probability values ≥ 0.9. Bootstrap values ≥ 70% for ML and MP are indicated above branches. Bootstrap values < 70% are indicated by *. Strains representing ex-type sequences are marked with ‘T’.

**Figure 8 jof-08-00214-f008:**
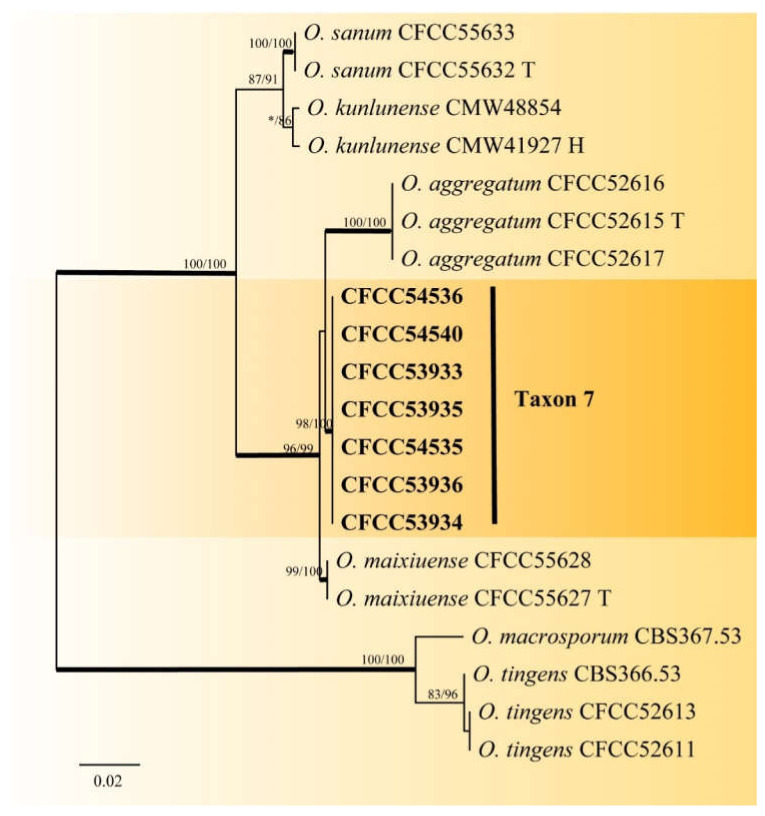
ML tree of Taxon 7 generated from the TUB2 sequence data. Novel sequences obtained in this study are presented in bold typeface. Bold branches indicate posterior probability values ≥ 0.9. Bootstrap values ≥ 70% for ML and MP are indicated above branches. Bootstrap values < 70% are indicated by *. Strains representing ex-type sequences are marked with ‘T’.

**Figure 9 jof-08-00214-f009:**
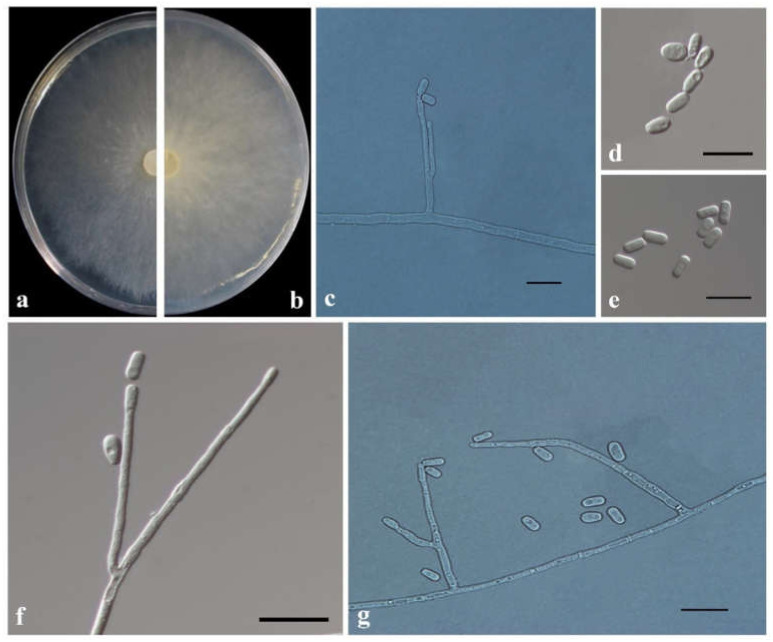
Morphological characteristics of *Graphilbum parakesiyea*. (**a**,**b**) Upper and reverse cultures on 2% MEA 8 days after inoculation. (**c**–**g**) Conidiogenous cells of hyalorhinocladiella-like asexual state and conidia. Scale bars: 10 μm (**c**–**g**).

**Figure 10 jof-08-00214-f010:**
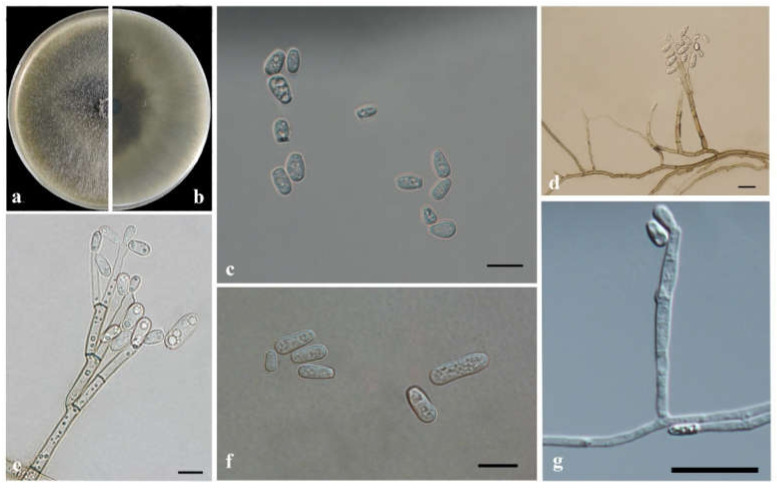
Morphological characteristics of *Leptographium qinlingense*. (**a**,**b**) Upper and reverse cultures on 2% MEA 5 days after inoculation; (**c**–**e**) Conidiogenous cells of leptographium-like asexual state and conidia; (**f**,**g**) Conidiogenous cells of hyalorhinocladiella-like asexual state and conidia. Scale bars: (**d**) = 20 μm; (**c**,**e**–**g**) = 10 μm.

**Figure 11 jof-08-00214-f011:**
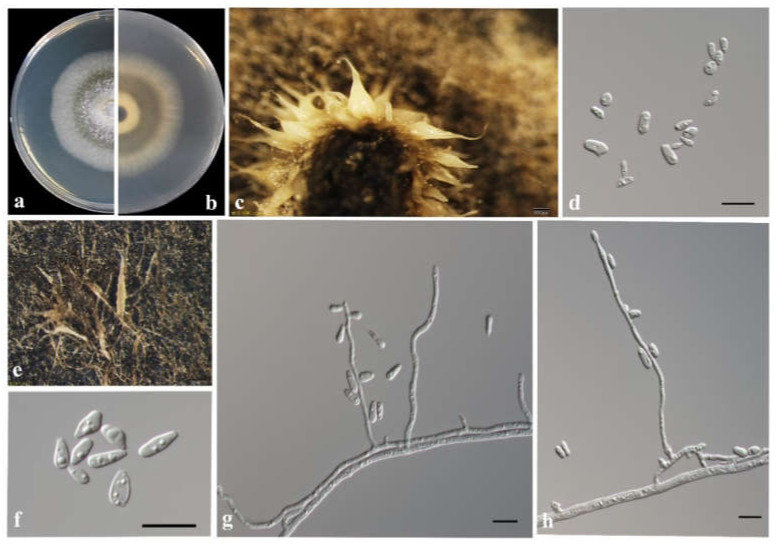
Morphological characteristics of *Ophiostoma shennongense*. (**a**,**b**) Upper and reverse cultures on 2% MEA 10 days after inoculation. (**c**,**e**) Brush-shaped conidiomata. (**d**,**f**–**h**) Conidiogenous cells of hyalorhinocladiella-like asexual state and conidia. Scale bars: (**c**,**e**) =20μm; (**d**,**f**–**h**) =10 μm.

**Table 1 jof-08-00214-t001:** Basic information on the sample collection plots and samples obtained from *P. armandii* infested *D. armandi* in western China.

Location	Longitude\Latitude	Altitude\m	No. of Hosts	No. of Tissue Pieces	No. of Adult Beetles/Galleries
Shennongjia Forest Area, Hubei Province	31°45′9″ N, 110°28′34″ E	1821	5	381	30
Dangchuan Forest Farm, Gansu Province	34°20′53″ N, 106°7′56″ E	1482	2	158	10
Foping county, Shaanxi Province	33°38′22″ N, 107°58′26″ E	1769	2	131	12
Huoditang Forest Form, Shaanxi Province	33°27′56″ N, 108°28′27″ E	2356	4	370	37

**Table 2 jof-08-00214-t002:** Representative strains of the ophiostomatoid fungi associated with *D. armandi* used for morphological and phylogenetic analysis and pathogenicity trials in this study.

Group Taxon	Name	Strain No.	Location	GenBank No.		
				LSU/ITS/ITS2-LSU	TUB2	TEF1-α
Taxon 1	*Esteya vermicola*	CFCC53942, CXY2518	Shennongjia Forest Area, Hubei Province	MW465992	MW690920	-
Taxon 2	*Graphilbum parakesiyea* sp. nov.	CFCC53924, CXY2516 T	Shennongjia Forest Area, Hubei Province	MW459985	MW770444	-
		CFCC54514, CXY2539	Foping County, Shaanxi Province	MW459986	MW770445	-
		CFCC54515, CXY2540	Foping County, Shaanxi Province	MW459987	MW770446	-
Taxon 3	*Graphium pseudormiticum*	CFCC53943, CXY2519	Shennongjia Forest Area, Hubei Province	MW459988	-	MW690919
Taxon 4	*Leptographium qinlingense*	CFCC53937, CXY2510	Dangchuan Forest Farm, Gansu Province	MW463377	MW723023	MW677124
		CFCC53938, CXY2511	Dangchuan Forest Farm, Gansu Province	MW463378	MW723024	MW677125
		CFCC53923, CXY2512	Dangchuan Forest Farm, Gansu Province	MW463379	MW723025	MW677126
		CFCC53939, CXY2513	Dangchuan Forest Farm, Gansu Province	MW463380	MW723026	MW677128
		CFCC53940, CXY2514	Dangchuan Forest Farm, Gansu Province	MW463381	MW723027	MW677129
		CFCC53941, CXY2515	Foping County, Shaanxi Province	MW463382	MW723028	MW677130
		CFCC54521, CXY2541	Foping County, Shaanxi Province	MW463383	MW723029	MW677131
		CFCC54516, CXY2542	Foping County, Shaanxi Province	MW463384	MW723030	MW677127
Taxon 5	*L. wushanense*	CFCC54524, CXY2543	Huoditang Forest Farm, Shaanxi Province	MW463385	MW690921	-
		CFCC54525, CXY2544	Huoditang Forest Farm, Shaanxi Province	MW463386	MW690922	-
		CFCC54526, CXY2545	Huoditang Forest Farm, Shaanxi Province	MW463387	MW690923	-
Taxon 6	*Ophiostoma shennongense* sp. nov.	CFCC53921, CXY2501 T	Shennongjia Forest Area, Hubei Province	MW459989	MW741822	-
		CFCC53922, CXY2502	Shennongjia Forest Area, Hubei Province	MW459990	MW741823	-
		CFCC53931, CXY2503	Dangchuan Forest Farm, Gansu Province	MW459991	MW741824	-
		CFCC53932, CXY2505	Foping County, Shaanxi Province	MW459992	MW741825	-
		CFCC54528, CXY2534	Dangchuan Forest Farm, Gansu Province	MW459993	MW741826	-
		CFCC54534, CXY2535	Huoditang Forest Farm, Shaanxi Province	MW459994	MW741827	-
		CFCC54533, CXY2536	Huoditang Forest Farm, Shaanxi Province	MW459995	MW741828	-
Taxon 7	*Ophiostoma* sp. 1	CFCC53933, CXY2506	Shennongjia Forest Area, Hubei Province	MW459996	MW759042	-
		CFCC53934, CXY2507	Shennongjia Forest Area, Hubei Province	MW459997	MW759043	-
		CFCC53935, CXY2508	Shennongjia Forest Area, Hubei Province	MW459998	MW759044	-
		CFCC53936, CXY2509	Shennongjia Forest Area, Hubei Province	MW459999	MW759045	-
		CFCC54535, CXY2531	Foping County, Shaanxi Province	MW460000	MW759046	-
		CFCC54536, CXY2532	Foping County, Shaanxi Province	MW460001	MW759047	-
		CFCC54540, CXY2533	Huoditang Forest Farm, Shaanxi Province	MW460002	MW759048	-

CFCC: China Forestry Culture Collection Center, Beijing, China; _CXY (Culture Xingyao): Culture collection of the Research Institute of Forest Ecology, Environment and Protection, Chinese Academy of Forestry. Sequences missing data are indicated by [-]; T = ex-holotype strains.

**Table 3 jof-08-00214-t003:** Strain numbers and percentage of various ophiostomatoid fungi isolated from *D. armandi* and their galleries in western China.

Taxon	Fungi Species	No. of Isolates 2018		No. of Isolates 2019		Total No. Strains	Percentage(%)
		Shennongjia Forest Area	Dangchuan Forest Farm	Foping County	Huoditang Forest Farm		
Taxon 1	*Esteyea vermicola*	1	0	0	0	1	0.14
Taxon 2	*Graphilbum parakesiyea*	2	0	15	8	25	3.60
Taxon 3	*Graphium pseudormiticum*	1	0	0	0	1	0.14
Taxon 4	*Leptographium qinlingensis*	2	24	31	5	62	8.92
Taxon 5	*L. wushanense*	0	0	0	34	34	4.89
Taxon 6	*Ophiostoma shennongensis*	266	42	27	215	549	78.99
Taxon 7	*Ophiostoma* sp. 1	6	3	4	9	22	3.17
	Total no. strains	280	69	77	269	695	

## Data Availability

All sequence data are available in NCBI GenBank following the accession numbers in the manuscript.

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
