# Peer review of "Diversity of Ophiostomatoid Fungi Associated with *Dendroctonus armandi* Infesting *Pinus armandii* in Western China"

_jof, 2022, doi:10.3390/jof8030214_

Round 1

Reviewer 1 Report

Several questions regarding the material:

  1. What were characteristics of each of those six sites/forest stands? E.g. age, species, origin, health status, etc.
  2. Characteristics of trees from which insects were collected: age, dbh, vigorous, recently attacked, dying …?
  3. How many trees sampled per site? From which part of a stem, approx. height.
  4. How many galleries sampled / insects collected per tree?
  5. How the insects and gallery samples were collected, e.g. removing bark, cutting off wood with sterilised knife (for galleries), picking beetles / wood samples with forceps pre-sterilised before each sampling?
  6. In case if beetles / galleries were collected from the same tree stem, what was approx. distance between those? Neighbouring or how distant beetles / galleries chosen, or?

Other remarks:

  1. Fig. 1a. The box at the bottom right corner is unclear, and no scale bar provided.

Author Response

Thank you very much for your attention to our manuscript. We are very grateful to you for your constructive comments and suggestions. We have carefully revised the manuscript (Manuscript ID: jof-1598854) accordingly, earnestly responded to all questions and corrected all relevant text according to the reviewers’ comments.

And all revisions made to the manuscript have be marked up using the “Track Changes” function. In the enclosed text, we present our response to the comments made by the reviewers.

Finally, we have resubmitted the revised manuscript in the review forum. We hope the revision will meet the standard of JoF and look forward to receiving your responses.

Best regards.

Yours sincerely,

Quan Lu on behalf of all the coauthors

E-mail: luquan@caf.ac.cn

Response to Reviewer 1 Comments

Point 1: What were characteristics of each of those six sites/forest stands? E.g. age, species, origin, health status, etc.

Response 1: We are very sorry that we have not clearly described the characteristics of those sites/forest stands in our study. All four sites are pure forests of Pinus armandii. Except for the P. armandii infested by Dendroctonus armandi, others are healthy, with a tree age of approximately 40 years old and diameter of approximately 40 to 60 cm.

We have added this detail into the revised manuscript. Please check in lines 71-73 at page 2 of the revised manuscript.

Point 2: Characteristics of trees from which insects were collected: age, dbh, vigorous, recently attacked, dying …?

Response 2: Thank you for your comments. All the trees used in this study showed signs of dead or dying (Figure 1d). Please check in line 73 at page 2 of the revised manuscript. For other characteristics, please see the response to the point 1.

Point 3: How many trees sampled per site? From which part of a stem, approx. height.

Response 3: Thank you very much. Please check in Table 1.

All D. armandi adults and their galleries collected were distributed as evenly as possible on the stem of P. armandii infested by D. armandi.

Point 4: How many galleries sampled / insects collected per tree?

Response 4: Thank you very much. Each P. armandii collected about 5-10 D. armandi adults and their galleries.

Point 5: How the insects and gallery samples were collected, e.g. removing bark, cutting off wood with sterilised knife (for galleries), picking beetles / wood samples with forceps pre-sterilised before each sampling?

Response 5: Thank you very much. The samples were collected by first removing bark with a hand saw and screwdriver, and then the beetles were individually placed in sterilized Eppendorf tubes using tweezers pre-sterilized, while their galleries were placed in sterile envelopes using sterilized knife. Please check in lines 73-75 at page 2 of the revised manuscript.

Point 6: In case if beetles / galleries were collected from the same tree stem, what was approx. distance between those? Neighbouring or how distant beetles / galleries chosen, or?

Response 6: Please see the response to the point 3.

Other remarks:

Point 7: Fig. 1a. The box at the bottom right corner is unclear, and no scale bar provided.

Response7: Thank you very much. We have added scale bar into the revised manuscript. Please check in Figure 1a. of the revised manuscript.

Thank you very much for your attention to our manuscript. We are very grateful to you for your constructive comments and suggestions. We have carefully revised the manuscript (Manuscript ID: jof-1598854) accordingly, earnestly responded to all questions and corrected all relevant text according to the reviewers’ comments.

And all revisions made to the manuscript have be marked up using the “Track Changes” function. In the enclosed text, we present our response to the comments made by the reviewers.

Finally, we have resubmitted the revised manuscript in the review forum. We hope the revision will meet the standard of JoF and look forward to receiving your responses.

Best regards.

Yours sincerely,

Quan Lu on behalf of all the coauthors

E-mail: luquan@caf.ac.cn

Response to Reviewer 1 Comments

Point 1: What were characteristics of each of those six sites/forest stands? E.g. age, species, origin, health status, etc.

Response 1: We are very sorry that we have not clearly described the characteristics of those sites/forest stands in our study. All four sites are pure forests of Pinus armandii. Except for the P. armandii infested by Dendroctonus armandi, others are healthy, with a tree age of approximately 40 years old and diameter of approximately 40 to 60 cm.

We have added this detail into the revised manuscript. Please check in lines 71-73 at page 2 of the revised manuscript.

Point 2: Characteristics of trees from which insects were collected: age, dbh, vigorous, recently attacked, dying …?

Response 2: Thank you for your comments. All the trees used in this study showed signs of dead or dying (Figure 1d). Please check in line 73 at page 2 of the revised manuscript. For other characteristics, please see the response to the point 1.

Point 3: How many trees sampled per site? From which part of a stem, approx. height.

Response 3: Thank you very much. Please check in Table 1.

All D. armandi adults and their galleries collected were distributed as evenly as possible on the stem of P. armandii infested by D. armandi.

Point 4: How many galleries sampled / insects collected per tree?

Response 4: Thank you very much. Each P. armandii collected about 5-10 D. armandi adults and their galleries.

Point 5: How the insects and gallery samples were collected, e.g. removing bark, cutting off wood with sterilised knife (for galleries), picking beetles / wood samples with forceps pre-sterilised before each sampling?

Response 5: Thank you very much. The samples were collected by first removing bark with a hand saw and screwdriver, and then the beetles were individually placed in sterilized Eppendorf tubes using tweezers pre-sterilized, while their galleries were placed in sterile envelopes using sterilized knife. Please check in lines 73-75 at page 2 of the revised manuscript.

Point 6: In case if beetles / galleries were collected from the same tree stem, what was approx. distance between those? Neighbouring or how distant beetles / galleries chosen, or?

Response 6: Please see the response to the point 3.

Other remarks:

Point 7: Fig. 1a. The box at the bottom right corner is unclear, and no scale bar provided.

Response7: Thank you very much. We have added scale bar into the revised manuscript. Please check in Figure 1a. of the revised manuscript.

Thank you very much for your attention to our manuscript. We are very grateful to you for your constructive comments and suggestions. We have carefully revised the manuscript (Manuscript ID: jof-1598854) accordingly, earnestly responded to all questions and corrected all relevant text according to the reviewers’ comments.

And all revisions made to the manuscript have be marked up using the “Track Changes” function. In the enclosed text, we present our response to the comments made by the reviewers.

Finally, we have resubmitted the revised manuscript in the review forum. We hope the revision will meet the standard of JoF and look forward to receiving your responses.

Best regards.

Yours sincerely,

Quan Lu on behalf of all the coauthors

E-mail: luquan@caf.ac.cn

Response to Reviewer 1 Comments

Point 1: What were characteristics of each of those six sites/forest stands? E.g. age, species, origin, health status, etc.

Response 1: We are very sorry that we have not clearly described the characteristics of those sites/forest stands in our study. All four sites are pure forests of Pinus armandii. Except for the P. armandii infested by Dendroctonus armandi, others are healthy, with a tree age of approximately 40 years old and diameter of approximately 40 to 60 cm.

We have added this detail into the revised manuscript. Please check in lines 71-73 at page 2 of the revised manuscript.

Point 2: Characteristics of trees from which insects were collected: age, dbh, vigorous, recently attacked, dying …?

Response 2: Thank you for your comments. All the trees used in this study showed signs of dead or dying (Figure 1d). Please check in line 73 at page 2 of the revised manuscript. For other characteristics, please see the response to the point 1.

Point 3: How many trees sampled per site? From which part of a stem, approx. height.

Response 3: Thank you very much. Please check in Table 1.

All D. armandi adults and their galleries collected were distributed as evenly as possible on the stem of P. armandii infested by D. armandi.

Point 4: How many galleries sampled / insects collected per tree?

Response 4: Thank you very much. Each P. armandii collected about 5-10 D. armandi adults and their galleries.

Point 5: How the insects and gallery samples were collected, e.g. removing bark, cutting off wood with sterilised knife (for galleries), picking beetles / wood samples with forceps pre-sterilised before each sampling?

Response 5: Thank you very much. The samples were collected by first removing bark with a hand saw and screwdriver, and then the beetles were individually placed in sterilized Eppendorf tubes using tweezers pre-sterilized, while their galleries were placed in sterile envelopes using sterilized knife. Please check in lines 73-75 at page 2 of the revised manuscript.

Point 6: In case if beetles / galleries were collected from the same tree stem, what was approx. distance between those? Neighbouring or how distant beetles / galleries chosen, or?

Response 6: Please see the response to the point 3.

Other remarks:

Point 7: Fig. 1a. The box at the bottom right corner is unclear, and no scale bar provided.

Response7: Thank you very much. We have added scale bar into the revised manuscript. Please check in Figure 1a. of the revised manuscript.

Reviewer 2 Report

Dear authors,

It is well designed study and really interesting. However I believe that you can improve some sections. I inserted a few comments in the manus,

Please describe all the micro morphological features of the included species.

You need to mention why you selected particular genes for different genera. It is very important. I would like to read the improved manus :)

Thank you

Author Response

Thank you very much for your attention to our manuscript. We are very grateful to you for your constructive comments and suggestions. We have carefully revised the manuscript (Manuscript ID: jof-1598854) accordingly, earnestly responded to all questions and corrected all relevant text according to the reviewers’ comments.

And all revisions made to the manuscript have be marked up using the “Track Changes” function. In the enclosed text, we present our response to the comments made by the reviewers.

Finally, we have resubmitted the revised manuscript in the review forum. We hope the revision will meet the standard of JoF and look forward to receiving your responses.

Best regards.

Yours sincerely,

Quan Lu on behalf of all the coauthors

E-mail: luquan@caf.ac.cn

Response to Reviewer 2 Comments

Point 1: L184 is there a particular reason for selecting genes for different analyses?

Did you follow any literatue to select the best genes for each genera

Response 1: Thank you for your comments. In ophiostomatoid fungi phylogenetic analyses, the LSU sequence is a suitable marker to infer the generic affinities of all ophiostomatoid fungi [1, 2], the ITS/ ITS2-LSU region would be useful to place strains within the complex, but the degree of polymorphism does not allow distinguishing species. The ITS region is more useful to place strains within Ophiostoma, Sporothrix, Graphilbum, Ceratocystiopsis and ITS2-LSU region is more useful to place strains within Leptographium, raffaelea [3, 4]. Usually, TUB2 and TEF1-αregions are better markers to identify and, where appropriate, to show the genetic diversity within ophiostomatoid fungi [1, 2, 5, 6].

Reference

  1. de Beer, Z. W.; Duong, T. A.; Wingfield, M. J. The divorce of Sporothrix and Ophiostoma: solution to a problematic relationship. Studies in mycology 2016, 83: 165–191.
  2. de Beer, Z. W.; Wingfield, M. J. Emerging lineages in the Ophiostomatales. In: KA Seifert, ZW de Beer, MJ Wingfield. The Ophiostomatoid Fungi: Expanding Frontiers. Utrecht, The Netherlands 2013, CBS 21–46.
  3. White, T. J.; Bruns, T.; Lee, S.; Taylor, J. Amplification and direct sequencing of fungal ribosomal RNA genes for phylogenetics. In: Innis MA, Gelfand DH, Sninsky, J. J.; White, T. J. (Eds) PCR protocols: a guide to methods and application. Academic Press, San Diego, USA1990, 315-322.
  4. Marincowitz, S.; Duong, T. A.; Taerum, S. J.; De Beer, Z. W.; Wingfield, M. J. Fungal associates of an invasive pine-infesting bark beetle, Dendroctonus valens, including seven new Ophiostomatalean fungi. Persoonia-Molecular Phylogeny and Evolution of Fungi 2020, 45(1), 177-195.
  5. Wang, H. M.; Wang, Z.; Liu, F.; Wu, C. X.; Zhang, S. F.; Kong, X. B.; Lu, Q.; Zhang, Z. Differential patterns of ophiostomatoid fungal communities associated with three sympatric Tomicus species infesting pines in south-western China, with a description of four new species. MycoKeys 2019, 50, 93.
  6. Zipfel, R. D.; de Beer, Z. W.; Jacobs, K. Multigene phylogenies define Ceratocystiopsis and Grosmannia distinct from Ophiostoma. Studies in Mycology 2006, 55: 75–97. 

Point 2: L221 in the diagram, correct the word lata

Response 2: Thank you very much. According to your suggestion, we have revised the words. Replace “lata” with “lato”. Please check in Figure 2.

Point 3: L236 (in Graphilbum s. str. )

Response 3: Thank you very much. According to your suggestion, we have revised the words. Replace “Taxon 2” with “Graphilbum s. str.”. Please check in line 235 at page 10 of the revised manuscript.

Point 4: L236 why did you use only this gene?

Response 4: Thank you for your comments. In this study, both ITS and TUB2 gene regions were used to identify genetic diversity within Graphilbum s. str., and Taxon 2 was represented by 3 sequences formed a well-supported, distinct clade.

Point 5: Replace “state” with “morph”, replace “Hyalorhinocladiella-like”, “Leptographium-like”; “Pesotum-like” with “hyalorhinocladiella-like”, “leptographium-like”, “pesotum-like”; replace “conidiogenous”, “conidia” with “Conidiogenous”, “Conidia”.

Response 5: Thank you very much. According to your suggestion, we have revised the words throughout the manuscript.

Point 6: 283, 383, Delete culture, holo,

Response 6: According to your suggestion, the words were deleted. Please check in lines 283 at page 15, 388 at page 18 of the revised manuscript.

Point 7: L287, you must describe the morphology of conidiogenous cells

L333, what about other characters

L387, please mention about other characters.

Response 7: We are very sorry that we have not clearly described the asexual morph of hyalorhinocladiella-like in our study. Referring to the publications of new species of ophiostomatoid fungi, descriptions of hyalorhinocladiella-like usually include the microscopic morphological characteristics of conidiogenous cells and conidia [1-3]. We have modified the description of hyalorhinocladiella-like. Please check in Morphology and taxonomy.

  1. Chang, R.; Duong, T. A.; Taerum, S. J.; Wingfield, M. J.; Zhou, X.; Yin, M.; De Beer, Z. W. Ophiostomatoid fungi associated with the spruce bark beetle Ips typographus, including 11 new species from China. Persoonia-Molecular Phylogeny and Evolution of Fungi 2019, 42(1), 50-74.
  2. Jankowiak, R.; Bilański, P.; Strzałka, B.; Linnakoski, R.; Bosak, A.; Hausner, G. Four new Ophiostoma species associated with conifer-and hardwood-infesting bark and ambrosia beetles from the Czech Republic and Poland. Antonie van Leeuwenhoek 2019, 112(10), 1501-1521.
  3. Wang, Z.; Zhou, Q.; Zheng, G.; Fang, J.; Han, F.; Zhang, X.; Lu, Q. Abundance and diversity of ophiostomatoid fungi associated with the Great Spruce Bark Beetle (Dendroctonus micans) in the Northeastern Qinghai-Tibet Plateau. Frontiers in microbiology 2021, 3082.

Point 8: L301, add figure number

Response 8: Thanks for your suggestions. The figure number was added. Please check in line 303 at page 16 of the revised manuscript.

Round 2

Reviewer 2 Report

Thank you for following the comments except morpho description. It is essential include the morpho description when you introduce new species. I highly recommend you to add those in your description.

Author Response

Dear reviewer, according to your suggestion, we have added morpho description in our manuscript. Please check in Morphology and taxonomy.

Point: L287, you must describe the morphology of conidiogenous cells

L333, what about other characters

L387, please mention about other characters.

Response: Thank you very much. We are very sorry that we have not clearly described the asexual morph of hyalorhinocladiella-like in our study. We have carefully consulted the description of hyalorhinocladiella-like in the articles of ophiostomatoid fungi, and modified the description of the conidiophore, conidiogenous cells and conidia characters of hyalorhinocladiella-like in detail. Please check in lines 285-291 at page 16, 339-345 at pages 17-18, 394-402 at page 19 of the revised manuscript.